# Relationships between radiation risk perception and health anxiety, and contribution of mindfulness to alleviating psychological distress after the Fukushima accident: Cross-sectional study using a path model

Yuya Kashiwazaki *, Yoshitake Takebayashi, Michio Murakami

Department of Health Risk Communication, Fukushima Medical University School of Medicine, Fukushima City, Japan

* nqi48911@fmu.ac.jp

**Data Availability Statement:** All relevant data are within the manuscript and its Supporting Information file.

## Abstract

One of biggest public health impacts of the Fukushima Daiichi Nuclear Power Station accident is psychosocial. Anxiety about radiation is still present, and radiation risk perception, particularly with regard to genetic effects, is known to affect mental health. However, roles of other risk factors such as health anxiety and of mindfulness remain to be proved. Here, we examined how radiation risk perception (genetic effects) mediates in health anxiety and psychological distress, and how mindfulness influences those variables. Seven years after the accident, we commissioned a self-reported online survey with 832 participants, 416 each from Fukushima and Tokyo, and modeled the relationship between those variables using Structural Equation Modeling. Health anxiety had a much stronger influence on psychological distress than radiation risk perception. Mindfulness was significantly correlated with both health anxiety and psychological distress, but not with radiation risk perception. The total effects on psychological distress were −0.38 by mindfulness and +0.38 by health anxiety. These results suggest the potential application of mindfulness-based interventions to alleviate health anxiety and psychological distress rather than therapy focused on radiation anxiety. The results underline the effectiveness of community support efforts in Fukushima and highlight the importance of enhancing mindfulness during the chronic phase following a disaster.

## Introduction

As a result of the Great East Japan Earthquake of 11 March 2011 and the subsequent nuclear power station accident, radionuclides diffused over much of Fukushima Prefecture, increasing anxiety about radiation. With consequent changes in living environment due to evacuation,

**Funding:** MM received JSPS KAKENHI grant number JP16H05894 to conduct the research. URL:https://www.jsps.go.jp/english/index.html The funders had no role in study design, data collection and analysis, decision to publish, or preparation of the manuscript.

**Competing interests:** The authors have declared that no competing interests exist.

multiple risks to health such as lifestyle-related diseases, poorer mental health, and substance misuse have increased [1,2]. Psychosocial effects caused greater concern than direct health effects resulting from radiation exposure [3]. Studies following both the Chernobyl and Fukushima accidents consistently showed that an individual's perception of the health effects of radiation ("radiation risk perception") is related to psychological distress and stigma [4–6], which complicated community and personal relations after each accident. Previous studies have shown that geographical conditions, such as distance from a nuclear power station and whether or not an area was under an evacuation order, influence perceptions of radiation risk [7,8]. Among residents of the Fukushima evacuation order areas, the annual Fukushima Health Management Survey (FHMS) found a mean psychological distress score of 6.3 (5.5 in men, 6.8 in women) in the fiscal year (FY; from April to March) 2011, and of 4.2 (4.0 in men, 4.4, in women) in FY 2017 [9]. Although it has steadily decreased, it remains high after 7 years and is thus a problem that requires a long-term perspective.

In order to respond to the public's anxiety and health problems, in parallel with health surveys, Japanese medical professionals have advised evacuees on managing radiation anxiety, provided general health consultations, and run comprehensive community support projects. The proportion of consultations related to radiation was low: 12.3% in FY 2012 and 4.4% in FY 2015 [10]. The topic of radiation was already regarded as taboo [10], few people wanted to talk about it actively, and the other health problems mentioned above were already of concern, so there was little demand for support related to radiation anxiety. Indeed, in the early period after the accident, when the health effects of low-dose exposure were unclear and there was a strong sense of distrust in the government's response, people were understandably uneasy. However, interdisciplinary research shows that genetic health effects of radiation exposure in Fukushima are not expected to occur at detectable levels [3]. Nevertheless, the background factors that maintain high levels of radiation risk perception are unclear.

Although how radiation risk perception affects psychological distress has not yet been fully elucidated, it is possible that a strong sense of anxiety about disorders and diseases, or health anxiety, may already exist in those with high radiation risk perception. Cognitive-behavioral models of hypochondriasis and health anxiety assume that a tendency to focus on information about health threats and bodily sensations contributes to its maintenance and deterioration [11–13]. The general public considers radiation risk to be much higher than experts measure it [14]. So those with high health anxiety are more likely to pay selective attention to elements of unease in rumors and local news than those with low health anxiety. In this way, health anxiety increases radiation risk perception. When it becomes so severe that it interferes with daily life, it is clinically defined as an illness anxiety disorder [15], which can range from mild to clinical [11,16]. The relationship between health anxiety and mental health (e.g., depression and anxiety) has been repeatedly reported [17], and health anxiety is treated with cognitive-behavioral therapy (CBT) [18]. As both general health anxiety and specific anxiety (such as radiation anxiety) affect mental health [19], the mode of support offered depends on which type of anxiety is stronger.

Mindfulness is offered as an intervention to support communities after disasters. Mindfulness refers to meditation and the mental state achieved through it, which is usually defined as the state of being attentive to and aware of what is occurring in the present moment, in a nonjudgmental or accepting way [20,21]. The introduction of mindfulness to treatment began when Kabat-Zinn demonstrated a therapeutic effect in a group intervention for patients with chronic diseases [21], and was later introduced into the context of CBT in psychotherapy through the systematization of Mindfulness-Based Cognitive Therapy (MBCT) [22]. Unlike previous approaches to correcting irrational beliefs in CBT, the purpose of MBCT is to objectively observe the thoughts and feelings that arise and to accept them as they are. Mindfulness-

based interventions have the advantage that they can be performed with a group and are useful in supporting communities. People with a high tendency toward mindfulness have been shown to be effective at reducing various mental and physical problems by promoting emotional self-regulation, reducing depressive symptoms and anxiety, and improving well-being [23,24]. If health anxiety is associated with post-accident psychological distress, mindfulness, by increasing flexibility of attention in the face of bias and persistence of attention [12], may have certain beneficial effects. A high tendency of mindfulness makes a person less susceptible to automatic processing [25–27]. In cognitive-behavioral models of health anxiety, automatic processing of health-related information, such as attention, are assumed as maintenance factors. Thus, mindfulness may reduce health anxiety and psychological distress through increased flexibility of attention. However, no studies have shown a relationship between mindfulness and radiation risk perception, and the role of health anxiety is not clear.

With the above in mind, the objectives of this study were to examine two points by modeling the effects of mindfulness, health anxiety, and radiation risk perception on psychological distress. First, we evaluated whether health anxiety or specific (radiation) anxiety has a stronger effect on psychological distress. Second, we examined the applicability of mindfulness as a means of providing support.

We proposed the following three hypotheses: 1. Radiation risk perception increases psychological distress. Because radiation risk in Fukushima residents is not relevant to Tokyo residents, this relationship is not present in the model of Tokyo residents. 2. Health anxiety increases both radiation risk perception and psychological distress. 3. Mindfulness reduces health anxiety, radiation risk perception, and psychological distress. This is the first study to reveal how health anxiety and risk perceptions are associated with psychological distress and how mindfulness contributes to these psychological responses.

## Methods

### Participants

Ethics approval for the study was granted by the Fukushima Medical University Ethics Committee (authorization No. General 30016). Residents of Fukushima, the disaster area, and Tokyo, Japan's capital city, were selected as study participants. A previous study reported on radiation risk perception (genetic effects) in Tokyo [28]. From a national perspective, levels of radiation anxiety in the Kanto regions (including Tokyo) were comparable to other regions, excluding Fukushima [29]. Participants who responded the online questionnaire had consented. An online survey was conducted with 832 previously registered participants, 416 each in Tokyo and Fukushima, by Macromill Co., Ltd. The survey was done on 25–26 August 2018 by members of the public aged 20 to 59 years. We asked Macromill to target participants whose sex and age ratios were in accordance with the demographics of each prefecture: Fukushima: men 52%, women 48%, 20s 19%, 30s 25%, 40s 27%, 50s 29%; Tokyo: men 51%, women 49%, 20s 22%, 30s 27%, 40s 29%, 50s 22%. Macromill invited panelists to fill out the questionnaire until enough had responded.

### Measures

Outcomes were measured on the 6-item Kessler scale (K6) [30] in the Japanese version [31]. The K6 consists of 6 questions about the degree of non-specific psychological distress (depression and anxiety) during the previous 30 days, each on a scale from 0 (none) to 4 (all the time), out of a total of 24 points.

Health anxiety was measured on a health anxiety inventory (HAI-J) designed for Japanese respondents [32] with reference to the Short Health Anxiety Inventory (SHAI) [33]. The

HAI-J consists of 10 items related to anxiety about health in the past 6 months and 4 items related to feelings at a time of serious illness, each on a scale from 1 (strongly disagree) to 4 (strongly agree). The SHAI can be used as a two- or three-factor structure [13,33]; the HAI-J used here has a three-factor structure: "Worry for physical health", "Negative cognition for serious illness", and "Hypochondriac tendencies for health". Only total scores were used. As in the previous study, we divided the total score of all items by the number of items for analysis (14) as the total score [32].

The Japanese version of the Five-Facet Mindfulness Questionnaire (FFMQ) [34,35] was used to evaluate dispositional mindfulness. The FFMQ was developed by reconstructing the five recognized measures of mindfulness by joint factor analysis, and is used in many studies as a gold standard for measuring mindfulness. The FFMQ includes 39 items, each rated on a 5-point scale from 1 (never or very rarely true) to 5 (very often or always true). It consists of five factors: "Observing", "Describing", "Acting with Awareness", "Non-judging", "Non-reactivity". Only total scores were used.

The perception of genetic risk due to radiation was assessed by a single question [36], because it is the most sensitive to psychological distress among risk perception indicators [5]. The question was "What do you think is the likelihood that the health of future (i.e., as-yet unborn) children and grandchildren will be affected as a result of the current level of radiation exposure [in Fukushima]?" and was measured on a 4-point Likert scale ranging from "4, very likely" to "1, very unlikely". The clause "[in Fukushima]" was added for Tokyo participants. Hereinafter, we use "radiation risk perception" to mean "perception of genetic risk" unless otherwise mentioned.

We obtained the permission from the author of HAI-J [32] to use the scale. The other questionnaires used did not require consent.

In addition, we collected age, sex, highest level of education, occupation, marital status, household income, and mental illness history. There were no missing data. Data of each participant are shown in S1 Table.

## Data analysis

We used the $\chi^2$ test to examine regional differences in basic characteristics and measures, and calculated Pearson's correlation coefficient, arithmetic means, standard deviations, and Cronbach's alpha coefficient of the measures. We also used Structural Equation Modeling (SEM) [37] to model the effect of each explanatory variable on the outcome variable. Differences between the Fukushima and Tokyo models were examined. The maximum likelihood method was used for model creation, and all variables were observational. Mindfulness and radiation risk perception were not correlated, so the path between them was not drawn. Other paths were drawn assuming that health anxiety affects radiation risk perception. Model fit was evaluated with the chi-squared statistic ($\chi^2$), the comparative fit index (CFI), the standardized root mean residual (SRMR), and the root mean square error of approximation (RMSEA). A model is typically accepted as an adequate fit when CFI > 0.90, and SRMR and RMSEA < 0.08 [37]. All statistical procedures were performed in SPSS v. 25 and Amos v. 24 software with a 0.05 significance level.

## Results

The basic characteristics of the subjects of this study are shown separately for Fukushima and Tokyo (Table 1). The average values of K6 were 7.25 for Fukushima and 6.92 for Tokyo, with no significant difference. There were significant differences in highest education level,

**Table 1. Characteristics of participants.**

| | | Total | | Fukushima | | Tokyo | | P |
|---|---|---|---|---|---|---|---|---|
| Individual attributes: n (%) | | | | | | | | |
| Sex | | | | | | | | |
| | Men | 427 | (51.3) | 215 | (51.7) | 212 | (51.0) | |
| | Women | 405 | (48.7) | 201 | (48.3) | 204 | (49.0) | |
| Age | | | | | | | | |
| | 20s | 169 | (20.3) | 78 | (18.8) | 91 | (21.9) | |
| | 30s | 214 | (25.7) | 102 | (24.5) | 112 | (26.9) | |
| | 40s | 236 | (28.4) | 114 | (27.4) | 122 | (29.3) | |
| | 50s | 213 | (25.6) | 122 | (29.3) | 91 | (21.9) | |
| Highest level of education | | | | | | | | |
| | Junior or high school graduate | 255 | (30.6) | 185 | (44.5) | 70 | (16.8) | *** |
| | University etc. graduate | 577 | (69.4) | 231 | (55.5) | 346 | (83.2) | |
| Occupation | | | | | | | | |
| | Employee etc.[a] | 465 | (55.9) | 212 | (51.0) | 253 | (60.8) | * |
| | Self employed etc.[b] | 48 | (5.8) | 27 | (6.5) | 21 | (5.0) | |
| | Other[c] | 319 | (38.3) | 177 | (42.5) | 142 | (34.1) | |
| Marital status | | | | | | | | |
| | Married | 436 | (52.4) | 239 | (57.5) | 197 | (47.4) | ** |
| | Unmarried and separation | 396 | (47.6) | 177 | (42.5) | 219 | (52.6) | |
| Annual household income | | | | | | | | |
| | Less than 3,000,000 yen | 155 | (22.7) | 91 | (27.7) | 64 | (18.0) | *** |
| | 3,000,000 yen—5,999,999 yen | 252 | (36.8) | 129 | (39.3) | 123 | (34.6) | |
| | More than 6,000,000 yen | 277 | (40.5) | 108 | (32.9) | 169 | (47.5) | |
| Mental illness history | | | | | | | | |
| | Yes | 71 | (8.5) | 35 | (8.4) | 36 | (8.7) | |
| | No | 761 | (91.5) | 381 | (91.6) | 380 | (91.3) | |
| Measures: means (SD) | | | | | | | | |
| | K6 | 7.09 | (5.48) | 7.25 | (5.43) | 6.92 | (5.53) | |
| | FFMQ | 117.09 | (12.79) | 115.97 | (12.05) | 118.21 | (13.40) | * |
| | HAI-J | 2.46 | (0.49) | 2.48 | (0.49) | 2.44 | (0.49) | |
| | Radiation risk perception | 2.61 | (0.83) | 2.56 | (0.82) | 2.66 | (0.84) | |

*$P < 0.05$

**$P < 0.01$

***$P < 0.001$. SD: standard deviation.

a Company employee, civil servant, non-profit-organization employee, teacher, health professional, or other professional.

b Agriculture, forestry, and fisheries workers and other self-employed workers.

c Part-time or casual worker, working on the side, housewife/husband, university student, technical college student, junior college student, preparatory school student, jobless, retired, etc.

occupation, marital status, and annual household income between the regions. There was no difference in mental illness between the regions.

We calculated the arithmetic mean and SD of the measures, internal consistency reliability coefficients (Cronbach's alpha), and Pearson's correlation coefficients between measures (Table 2). Values of Cronbach's alpha were as follows: K6, 0.892; FFMQ, 0.786; HAI-J, 0.868. Correlations were negative between K6 and FFMQ ($r = -0.381$) and between FFMQ and HAI-J ($r = -0.181$); and positive between K6 and HAI-J ($r = 0.439$), between K6 and radiation

**Table 2. Arithmetic means, standard deviation (SD), Cronbach's alpha, and correlations of the measures.**

|  | 1 |  | 2 |  | 3 |  | Mean | SD | α |
|---|---|---|---|---|---|---|---|---|---|
| 1. K6 | - |  |  |  |  |  | 7.09 | 5.48 | 0.892 |
| 2. FFMQ | -0.381 | ** | - |  |  |  | 117.09 | 12.79 | 0.786 |
| 3. HAI-J | 0.439 | ** | -0.181 | ** | - |  | 2.46 | 0.49 | 0.868 |
| 4. Radiation risk perception | 0.162 | ** | -0.017 |  | 0.228 | ** | 2.61 | 0.83 | - |

$^{**}P < 0.01$.

risk perception ($r = 0.162$), and between HAI-J and radiation risk perception ($r = 0.228$). There was no significant correlation between FFMQ and radiation risk perception.

Table 3 shows correlations between measures in each region. The correlation between K6 and FFMQ was slightly stronger in Tokyo, while those between K6 and HAI-J and between K6 and radiation risk perception were slightly stronger in Fukushima.

Using the above results, we created path models of the effects of mindfulness, health anxiety, and radiation risk perception as explanatory variables on psychological distress (Fig 1A). The model of all participants had good a fit to the data: $\chi^2 = 4.689$, df = 7, CFI = 1.000, SRMR = 0.008, RMSEA < 0.001. All path coefficients were significant. The direct effect of radiation risk perception on psychological distress was 0.07, that of health anxiety was 0.37, and that of mindfulness was −0.31. The indirect effect of mindfulness via health anxiety was −0.07 (−0.18 × 0.37), and that of health anxiety via radiation risk perception was 0.02. The total effect of health anxiety on psychological distress was 0.38, and that of mindfulness was −0.38. In the model of Fukushima, the total effect of health anxiety on psychological distress was strong at 0.42, and that of mindfulness was modest at −0.34 than Tokyo (Fig 1B). In the model of Tokyo, the total effect of health anxiety on psychological distress was modest at 0.35, and that of mindfulness was strong at −0.42. Further, there was no significant effect of radiation risk perception on psychological distress (Fig 1C). There were no significant differences in any path coefficients between the regions. In addition, as a result of Table 1, regional differences in highest level of education were conspicuous, stratified path analysis was performed. A stratified model (regions × highest education level) showed a similar result (S1 Fig).

## Discussion

The aims of this study were to investigate the relationships among mindfulness, health anxiety, radiation risk perception, and psychological distress and to examine the applicability of mindfulness for support. The K6 score was 7.09, higher than the mean score of 4.2 in the 2017 Fukushima Health Management Survey targeting the evacuees [9]. This increase may be due to the differences in survey methods, or in the varying characteristics of panelists registered

**Table 3. Correlations of the variables by region.**

|  | 1 |  | 2 |  | 3 |  | 4 |  |
|---|---|---|---|---|---|---|---|---|
| 1. K6 | - |  | -0.337 | ** | 0.471 | ** | 0.203 | ** |
| 2. FFMQ | -0.417 | ** | - |  | -0.213 | ** | -0.053 |  |
| 3. HAI-J | 0.406 | ** | -0.146 | ** | - |  | 0.218 | ** |
| 4. Radiation risk perception | 0.127 | ** | 0.005 |  | 0.244 | ** | - |  |

$^{**}P < 0.01$.

Upper right (italics), Fukushima; lower left, Tokyo.

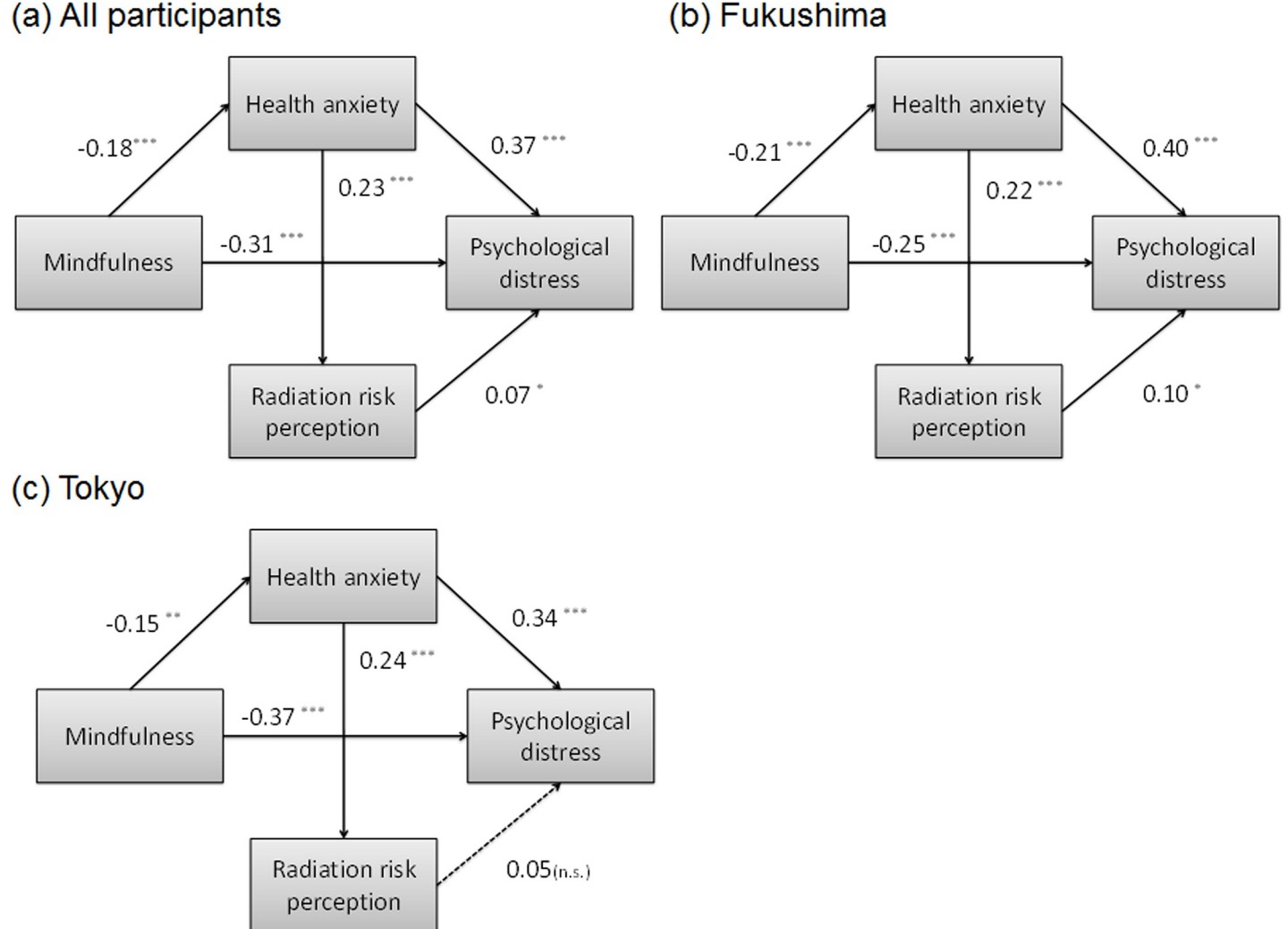

**Fig 1.** Path model results for (a) all participants, (b) Fukushima, and (c) Tokyo by Structural Equation Modeling. All the coefficients are standardized estimates, $^*P < 0.05$, $^{**}P < 0.01$, $^{***}P < 0.001$.

with the survey company. First, concerning the differences in survey methods, the FHMS was mainly conducted by mail, while this study was conducted via an online survey [38]. Second, the FHMS was targeted at evacuees, while this study examined a sample of users registered with an online survey company throughout the whole of the Fukushima Prefecture. Third, while the FHMS had a high proportion of elderly respondents (average 62.4 years), this study sampled sex and age ratios according to the demographics of each prefecture. The mean FFMQ score was 117.09, close to that of 113.19 in previous research in Japan [35]. Similarly, the mean score of HAI-J was 2.46, similar to that of 2.71 in previous research in Japan [32].

SEM analysis showed that radiation risk perception had no effect on psychological distress in Tokyo, which is a reasonable result. Its effect in Fukushima was small, whereas the effect of health anxiety was strong. So regardless of region, the apparent relationship between radiation risk perception and psychological distress was explained by the strong relationship between health anxiety and psychological distress (Fig 1). In other words, these indicate that both general health anxiety and specific anxiety affect mental health, and that general health anxiety has

a stronger effect on mental health. Although the previous study reported that specific anxiety regarding cancer more strongly affects mental health, compared to general health anxiety [19], the relative magnitude of the effect may vary depending on the type of specific anxiety and mental health-measured content.

A review of radiation risk perception studies in Fukushima showed that safety behavior (e.g., seeking information about radiation, checking dose rates, and staying out of high-dose areas) was related to high radiation risk perception [39]. Checking behavior and seeking reassurance (safety behaviors) contribute to the maintenance of health anxiety [11]. Our finding (i.e., health anxiety increases radiation risk perception and psychological distress) is consistent with those previous studies. Safety behavior is used to escape emotional distress by seeking reassurance in response to anxiety or worry caused by a threat. However, when it is unhelpful, CBT shows that it is only a temporary coping strategy, and basic anxiety is maintained [40]. In addition, health anxiety is also characterized by intolerance of uncertainty, and in most cases increases the perception of threats, because safety behavior cannot obtain complete certainty that leads to reassurance. That is, unhelpful safety behavior maintains selective attention to threats and thus continues to cause high radiation risk perception and psychological distress.

Our results also demonstrated strong direct effects of mindfulness on psychological distress, moderate direct effects of it on health anxiety, and overall mitigation of psychological distress. These effects are consistent with previous reports that showed the positive effects of mindfulness on health anxiety [41,42] and psychological distress [43,44]. This relationship suggests that somatosensory amplification and misinterpretation of bodily sensations can aggravate health anxiety, and so mindfulness (including paying attention to bodily sensations and accepting them without interpretation or value judgment) may have been effective in reducing health anxiety. People who have health anxiety respond to negative automatic thoughts and expand their negative interpretations by worry, rumination, and a focus on threats. Qualitative intervention studies using MBCT have reported reduced automatic responses to heightened bodily sensations and thoughts and increased awareness of the relationship between physical sensations and thoughts and anxiety, but lower effects on safety-seeking behaviors [41]. Increasing mindfulness may suppress responses to these particularly cognitive aspects. In practice, therefore, it may be more effective to approach health anxiety in terms of reducing psychological distress than to focus on radiation anxiety. In fact, Imamura et al. showed that a mental health promotion program using behavioral-activation-based intervention also reduced psychological distress without changing radiation risk perception [45]. This result supports the importance of post-accident community support efforts aimed at general health problems, not just radiation anxiety. It also suggests that the implementation of programs such as mindfulness courses as a part of community support could contribute to the improvement of general health as well as the reduction of health anxiety.

This study had some limitations. First, it was based on an online survey. The higher average K6 score than that of the Fukushima Health Management Survey conducted by mail suggests the possibility of selection bias due to differences in survey methods. Second, it excluded people over 60 years of age, because older internet users may not be representative of older residents in general. As radiation risk perception is known to depend on age and is higher in older people [39], caution should be exercised in generalizing the conclusions of this study. Third, this is a cross-sectional study based on a self-reported questionnaire. The theoretical framework used in modeling assumed causal relationships. It will be necessary to verify actual effects through future intervention studies. Fourth, since this study was conducted about seven years after the accident, it may not be applicable to the acute phase after an accident.

However, the results point to a relationship between psychological distress and radiation risk perception that can be explained by the link between psychological distress and health

anxiety. The results support the use of mindfulness as an option for post-disaster community support. These findings are meaningful in terms of recovery in both Fukushima global preparedness for future disasters.

## Conclusions

Nuclear accidents raise public concerns about the health effects of radiation, which negatively affects the mental health of those affected. However, it was not clear how radiation risk perception leads to psychological distress. In addition, effective support methods had not been established. We showed that the apparent relationship between radiation risk perception and psychological distress can be explained by the strong relationship between health anxiety and psychological distress. Mindfulness can reduce health anxiety and psychological distress.

## Supporting information

**S1 Fig. Pass model results for stratification of region and education level.** (a) Lower-educated in Fukushima, (b) Highly-educated in Fukushima, (c) Lower-educated in Tokyo, (d) Highly-educated in Tokyo.
(DOCX)

**S1 Table. Data of each participant.**
(XLSX)

## Acknowledgments

We would like to thank all of the participants in answering the questionnaire and the FMU colleague who supported us with useful discussion in this study.

## Author Contributions

**Conceptualization:** Yuya Kashiwazaki, Yoshitake Takebayashi, Michio Murakami.

**Formal analysis:** Yuya Kashiwazaki.

**Funding acquisition:** Michio Murakami.

**Methodology:** Yuya Kashiwazaki, Yoshitake Takebayashi, Michio Murakami.

**Supervision:** Michio Murakami.

**Visualization:** Yuya Kashiwazaki.

**Writing – original draft:** Yuya Kashiwazaki.

**Writing – review & editing:** Yoshitake Takebayashi, Michio Murakami.

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
