## [Decision Letter · Decision Letter 0]

28 Apr 2020

PONE-D-20-07222

Relationships between radiation risk perception and health anxiety, and contribution of mindfulness to alleviating psychological distress after the Fukushima accident: Cross-sectional study using a path model

PLOS ONE

Dear Mr. Kashiwazaki,

Thank you for submitting your manuscript to PLOS ONE. After careful consideration, we feel that it has merit but does not fully meet PLOS ONE’s publication criteria as it currently stands. Therefore, we invite you to submit a revised version of the manuscript that addresses the points raised during the review process.

The two reviewers several major and minor concerns about your manuscript. Please revise your manuscript carefully.

We would appreciate receiving your revised manuscript by Jun 12 2020 11:59PM. To enhance the reproducibility of your results, we recommend that if applicable you deposit your laboratory protocols in protocols.io, where a protocol can be assigned its own identifier (DOI) such that it can be cited independently in the future. For instructions see: http://journals.plos.org/plosone/s/submission-guidelines#loc-laboratory-protocols

We look forward to receiving your revised manuscript.

Kind regards,

Kenji Hashimoto, PhD

Academic Editor

PLOS ONE

Reviewers' comments:

Reviewer's Responses to Questions

**Comments to the Author**

1. Is the manuscript technically sound, and do the data support the conclusions?

Reviewer #1: Partly

Reviewer #2: Partly

2. Has the statistical analysis been performed appropriately and rigorously? 

Reviewer #1: I Don't Know

Reviewer #2: I Don't Know

3. Have the authors made all data underlying the findings in their manuscript fully available?

Reviewer #1: No

Reviewer #2: No

4. Is the manuscript presented in an intelligible fashion and written in standard English?

Reviewer #1: Yes

Reviewer #2: Yes

5. Review Comments to the Author

Reviewer #1: Analyzing a self-reported online survey data, the authors investigated the relationships between radiation risk perception, health anxiety, mindfulness, and psychological distress. They used Structural Equation Modeling to specify the relationship between those variables. Result showed that health anxiety had a much stronger influence on psychological distress than radiation risk perception. Mindfulness was significantly correlated with both health anxiety and psychological distress, but not with radiation risk perception. The total effects on psychological distress were −0.38 by mindfulness and +0.38 by health anxiety. The authors suggested the potential application of mindfulness-based interventions to alleviate health anxiety, psychological distress, and radiation anxiety.

The findings will be of interest to researchers in the field.

I have the following concerns.

#1 Abstract, Line 41, The authors state that “The results support the effectiveness of community support in Fukushima and highlight the importance of enhancing mindfulness in emergencies” There is no data presented in the manuscript to support the effectiveness of mindfulness in emergencies. This sentence should be revised.

#2 Introduction, Line 82, “As general health anxiety and specific anxiety (such as radiation anxiety) independently affect mental health [17], the mode of support offered depends on which is stronger.”

I think this sentence is inconsistent with the present findings. An explanatory is necessary in Discussion section.

#3 Introduction, Line 107, “Therefore, this study had two objectives. First, we created a model for the effects of mindfulness, health anxiety, and radiation risk perception on psychological distress. Second, we examined the applicability of mindfulness for support.”

I think the authors created a model for the effects of mindfulness, health anxiety, and radiation risk perception on psychological distress to examine the applicability of mindfulness for support. It appears that creating a model for the effects of mindfulness, health anxiety, and radiation risk perception on psychological distress is a means, while examining the applicability of mindfulness for support is an aim. In short, I think this study had one objective.

#4 Introduction, Line 114, “Although existing knowledge is limited, we first explored the idea that mindfulness reduces radiation risk perception, as is it does with health anxiety.”

The authors proposed three hypothesis (line110) and they first explored the third hypothesis. What has become of the first and second hypotheses? If the three hypotheses are for path analysis and authors explored all three hypotheses together, that sentence is confusing.

#5 Discussion, Line 239, “The increase may be due to a difference in survey methods or in the characteristics of panelists registered with the survey company.”

For the reason mentioned above, I think the explanation should be supported by references.

In conclusion, although this is a valuable paper that presents the relationships between radiation risk perception, health anxiety, mindfulness a self-reported online survey data, and psychological distress using a self-reported online survey data, it requires moderate revision.

Reviewer #2: This study showed public health impacts of the Fukushima Daiichi Nuclear Power Station accident.

Authors stated that enhancing mindfulness is effective in emergencies like Nuclear Power Station accident.

They commissioned a self-reported online survey with 832 participants, 416 each from Fukushima and Tokyo, and modeled the relationship between those variables using Structural Equation Modeling.

The strength of this study is large number of participants.

However I have a few suggestions.

#1: Why did the authors compare between Tokyo residents and Fukushima residents? (Tokyo residents may have fear of radiation risk more than Okinawa residents.)

#2: Could you show us the data between Fukushima-city and Iwaki-city? (It may be different even in same prefecture.)

6. PLOS authors have the option to publish the peer review history of their article (what does this mean?). If published, this will include your full peer review and any attached files.

Reviewer #1: No

Reviewer #2: No

---

## [Author Response · Author response to Decision Letter 0]

7 Jun 2020

Reviewer 1

#1 Abstract, Line 41, The authors state that “The results support the effectiveness of community support in Fukushima and highlight the importance of enhancing mindfulness in emergencies” There is no data presented in the manuscript to support the effectiveness of mindfulness in emergencies. This sentence should be revised.

Response 1

Thank you so much for your comment. Accordingly, we have revised the abstract as bellow.

[Abstract, Line 39]

The results underline the effectiveness of community support efforts in Fukushima and highlight the importance of enhancing mindfulness during the chronic phase following a disaster.

#2 Introduction, Line 82, “As general health anxiety and specific anxiety (such as radiation anxiety) independently affect mental health [17], the mode of support offered depends on which is stronger.” I think this sentence is inconsistent with the present findings. An explanatory is necessary in Discussion section.

Response 2

Thank you so much for your suggestion. We have changed the explanation in introduction and added the explanation in Discussion. 

[Introduction, Line 85]

As both general health anxiety and specific anxiety (such as radiation anxiety) affect mental health [19], the mode of support offered depends on which type of anxiety is stronger.

[Discussion, Line 264]

In other words, these indicate that both general health anxiety and specific anxiety affect mental health, and that general health anxiety has a stronger effect on mental health. Although the previous study reported that specific anxiety regarding cancer more strongly affects mental health, compared to general health anxiety [19], the relative magnitude of the effect may vary depending on the type of specific anxiety and mental health-measured content.

#3 Introduction, Line 107, “Therefore, this study had two objectives. First, we created a model for the effects of mindfulness, health anxiety, and radiation risk perception on psychological distress. Second, we examined the applicability of mindfulness for support.” I think the authors created a model for the effects of mindfulness, health anxiety, and radiation risk perception on psychological distress to examine the applicability of mindfulness for support. It appears that creating a model for the effects of mindfulness, health anxiety, and radiation risk perception on psychological distress is a means, while examining the applicability of mindfulness for support is an aim. In short, I think this study had one objective.

Response 3

We agree with you. As you pointed out, model creation is a means. In addition, understanding radiation risk perception with the concept of health anxiety was also one of the major objectives, we added it.

[Introduction, Line 110]

With the above in mind, the objectives of this study were to examine two points by modeling the effects of mindfulness, health anxiety, and radiation risk perception on psychological distress. First, we evaluated whether health anxiety or specific (radiation) anxiety has a stronger effect on psychological distress. Second, we examined the applicability of mindfulness as a means of providing support.

#4 Introduction, Line 114, “Although existing knowledge is limited, we first explored the idea that mindfulness reduces radiation risk perception, as is it does with health anxiety.” The authors proposed three hypothesis (line110) and they first explored the third hypothesis. What has become of the first and second hypotheses? If the three hypotheses are for path analysis and authors explored all three hypotheses together, that sentence is confusing.

Response 4

Based on your suggestion, we have modified the text as follows.

[Introduction, Line 119]

This is the first study to reveal how health anxiety and risk perceptions are associated with psychological distress and how mindfulness contributes to these psychological responses.

#5 Discussion, Line 239, “The increase may be due to a difference in survey methods or in the characteristics of panelists registered with the survey company.” For the reason mentioned above, I think the explanation should be supported by references.

Response 5

We added to discussion about the difference between this study and FHMS.

[Discussion, Line 249]

This increase may be due to the differences in survey methods, or in the varying characteristics of panelists registered with the survey company. First, concerning the differences in survey methods, the FHMS was mainly conducted by mail, while this study was conducted via an online survey [38]. Second, the FHMS was targeted at evacuees, while this study examined a sample of users registered with an online survey company throughout the whole of the Fukushima Prefecture. Third, while the FHMS had a high proportion of elderly respondents (average 62.4 years), this study sampled sex and age ratios according to the demographics of each prefecture.

[References]

38. Yasumura S, Hosoya M, Yamashita S, Kamiya K, Abe M, Akashi M, et al. Study protocol for the Fukushima health management survey. J Epidemiol. 2012;22: 375–383. doi:10.2188/jea.JE20120105

 

Reviewer 2

#1: Why did the authors compare between Tokyo residents and Fukushima residents? (Tokyo residents may have fear of radiation risk more than Okinawa residents.)

Response 1

Thank you so much for your comment. The reason we chose Tokyo as a compare group is that it is a capital city, that is, a representative city in Japan. Besides, the previous study reported radiation risk perception (genetic effects) in Tokyo [28]. Because of the comparable results were available, Tokyo residents were chosen as the participants. Furthermore, from a national perspective, levels of radiation anxiety in the Kanto region including Tokyo were comparable to other regions excluding Fukushima. We have added the explanations as below. 

[Method, Line 126]

Residents of Fukushima, the disaster area, and Tokyo, Japan’s capital city, were selected as study participants. A previous study reported on radiation risk perception (genetic effects) in Tokyo [28]. From a national perspective, levels of radiation anxiety in the Kanto regions (including Tokyo) were comparable to other regions, excluding Fukushima [29].

[References]

28. Shirai K, Yoshizawa N, Takebayashi Y, Murakami M. Modeling reconstruction-related behavior and evaluation of influences of major information sources. Schnettler B, editor. PLoS One. 2019;14: e0221561. doi:10.1371/journal.pone.0221561

29. Sekiya N. Research Survey of Consumer Psychology about Radioactive Contaminationafter the Accident at TEPCO’s Fukushima Daiichi Nuclear Power Stations. J Soc Saf Sci. 2016;29: 143–153. doi:10.11314/jisss.29.143 [in Japanese]

#2: Could you show us the data between Fukushima-city and Iwaki-city? (It may be different even in same prefecture.)

Response 2

Unfortunately, we are not able to compare the results of this survey because we have not more detailed geographic information. However, previous studies that compared risk perception among Fukushima Prefecture regions showed that distance from a nuclear power station, and regions within the prefecture are one of the factors that influence risk perception. Furthermore, the previous report [28] showed the anxiety of Aizu-Wakamatsu City is lower than that of Fukushima City and Iwaki City and that a difference between Fukushima City and Iwaki City was small. Information on these existing studies was added to the Introduction section.

[Introduction, Line 53]

Previous studies have shown that geographical conditions, such as distance from a nuclear power station and whether or not an area was under an evacuation order, influence perception of radiation risk [7, 8].

[References]

7. Satoshi Suzuki, Michio Murakami, Tatsuhiro Nishikiori, Shigeki Harada: Annual changes in the Fukushima residents' views on the safety of water and air environments and their associations with the perception of radiation risks, Journal of Radiation Research, Supplement - Highlight Articles of the First International Symposium, 59(S2), pp.ii31-ii39, 2018. doi: 10.1093/jrr/rrx096.

8. Nakayama C, Sato O, Sugita M, Nakayama T, Kuroda Y, Orui M, et al. Lingering health-related anxiety about radiation among Fukushima residents as correlated with media information following the accident at Fukushima Daiichi Nuclear Power Plant. Seale H, editor. PLoS One. 2019;14: e0217285. doi:10.1371/journal.pone.0217285

In addition, data of each participant was added as a supporting information.

We believe that we have addressed your feedback and hope that these revisions persuade you to accept our submission. Thank you for your generous consideration.

---

## [Decision Letter · Decision Letter 1]

17 Jun 2020

Relationships between radiation risk perception and health anxiety, and contribution of mindfulness to alleviating psychological distress after the Fukushima accident: Cross-sectional study using a path model

PONE-D-20-07222R1

Dear Dr. Kashiwazaki,

We’re pleased to inform you that your manuscript has been judged scientifically suitable for publication and will be formally accepted for publication once it meets all outstanding technical requirements.

Kind regards,

Kenji Hashimoto, PhD

Section Editor

PLOS ONE

Additional Editor Comments (optional):

Reviewers' comments:

Reviewer's Responses to Questions

**Comments to the Author**

1. If the authors have adequately addressed your comments raised in a previous round of review and you feel that this manuscript is now acceptable for publication, you may indicate that here to bypass the “Comments to the Author” section, enter your conflict of interest statement in the “Confidential to Editor” section, and submit your "Accept" recommendation.

Reviewer #1: All comments have been addressed

Reviewer #2: All comments have been addressed

2. Is the manuscript technically sound, and do the data support the conclusions?

Reviewer #1: (No Response)

Reviewer #2: Yes

3. Has the statistical analysis been performed appropriately and rigorously? 

Reviewer #1: Yes

Reviewer #2: I Don't Know

4. Have the authors made all data underlying the findings in their manuscript fully available?

Reviewer #1: Yes

Reviewer #2: Yes

5. Is the manuscript presented in an intelligible fashion and written in standard English?

Reviewer #1: Yes

Reviewer #2: Yes

6. Review Comments to the Author

Reviewer #1: (No Response)

Reviewer #2: #1: Why did the authors compare between Tokyo residents and Fukushima residents? (Tokyo residents may have fear of radiation risk more than Okinawa residents.)

Response 1

Thank you so much for your comment. The reason we chose Tokyo as a compare group is that it is a capital city, that is, a representative city in Japan. Besides, the previous study reported radiation risk perception (genetic effects) in Tokyo [28]. Because of the comparable results were available, Tokyo residents were chosen as the participants. Furthermore, from a national perspective, levels of radiation anxiety in the Kanto region including Tokyo were comparable to other regions excluding Fukushima. We have added the explanations as below.

[Method, Line 126]

Residents of Fukushima, the disaster area, and Tokyo, Japan’s capital city, were selected as study participants. A previous study reported on radiation risk perception (genetic effects) in Tokyo [28]. From a national perspective, levels of radiation anxiety in the Kanto regions (including Tokyo) were comparable to other regions, excluding Fukushima [29].

[References]

28. Shirai K, Yoshizawa N, Takebayashi Y, Murakami M. Modeling reconstruction- related behavior and evaluation of influences of major information sources. Schnettler B, editor. PLoS One. 2019;14: e0221561. doi:10.1371/journal.pone.0221561

29. Sekiya N. Research Survey of Consumer Psychology about Radioactive Contaminationafter the Accident at TEPCO’s Fukushima Daiichi Nuclear Power Stations. J Soc Saf Sci. 2016;29: 143–153. doi:10.11314/jisss.29.143 [in Japanese]

#2: Could you show us the data between Fukushima-city and Iwaki-city? (It may be different even in same prefecture.)

=> Thank you so much for your sincere correspondence.

Response 2

Unfortunately, we are not able to compare the results of this survey because we have not more detailed geographic information. However, previous studies that compared risk perception among Fukushima Prefecture regions showed that distance from a nuclear

power station, and regions within the prefecture are one of the factors that influence risk perception. Furthermore, the previous report [28] showed the anxiety of Aizu-Wakamatsu City is lower than that of Fukushima City and Iwaki City and that a difference between Fukushima City and Iwaki City was small. Information on these existing studies was added to the Introduction section.

[Introduction, Line 53]

Previous studies have shown that geographical conditions, such as distance from a nuclear power station and whether or not an area was under an evacuation order, influence perception of radiation risk [7, 8].

[References]

7. Satoshi Suzuki, Michio Murakami, Tatsuhiro Nishikiori, Shigeki Harada: Annual changes in the Fukushima residents' views on the safety of water and air environments and their associations with the perception of radiation risks, Journal of Radiation

Research, Supplement - Highlight Articles of the First International Symposium, 59(S2), pp.ii31-ii39, 2018. doi:10.1093/jrr/rrx096.

8. Nakayama C, Sato O, Sugita M, Nakayama T, Kuroda Y, Orui M, et al. Lingering health-related anxiety about radiation among Fukushima residents as correlated with media information following the accident at Fukushima Daiichi Nuclear Power Plant. Seale H, editor. PLoS One. 2019;14: e0217285. doi:10.1371/journal.pone.0217285

In addition, data of each participant was added as a supporting information.

We believe that we have addressed your feedback and hope that these revisions persuade you to accept our submission. Thank you for your generous consideration.

=> Thank you so much for your sincere correspondence to my requests.

7. PLOS authors have the option to publish the peer review history of their article (what does this mean?). If published, this will include your full peer review and any attached files.

Reviewer #1: No

Reviewer #2: No

---

## [Editor Report · Acceptance letter]

24 Jun 2020

PONE-D-20-07222R1 

Relationships between radiation risk perception and health anxiety, and contribution of mindfulness to alleviating psychological distress after the Fukushima accident: Cross-sectional study using a path model 

Dear Dr. Kashiwazaki:

I'm pleased to inform you that your manuscript has been deemed suitable for publication in PLOS ONE. Congratulations! Your manuscript is now with our production department. 

Kind regards, 

on behalf of

Prof. Kenji Hashimoto 

Section Editor

PLOS ONE